

# Association between sports expertise and visual attention in male and female soccer players

Peng Jin[1,2], Zheqi Ji[1], Tianyi Wang[3] and Xiaomin Zhu[4]

[1] Department of Physical Education, Southeast University, Nanjing, China
[2] Southeast university research institute of sports science, Nanjing, China
[3] College of Physical education and health, East China Normal University, Shanghai, China
[4] Department of Physical education, Nanjing University of Aeronautics & Astronautics, Nanjing, China

Corresponding author
Xiaomin Zhu, 280092110@qq.com

## ABSTRACT

**Background.** Visual attention plays a crucial role in daily living and in sports, affecting an athlete's performance and thus, potentially, the outcome of a match. However, studies assessing the association between the level of sports expertise and visual attention have yielded mixed results. This study was conducted to examine whether visual attention could be developed with increased sports expertise, and whether visual attention differed between male athletes and female athletes.

**Methods.** A total of 128 participants were included in this study: 64 first-level national soccer athletes recruited from college soccer teams (considered elite athletes; 32 men and 32 women with similar soccer performance requirements and training experience), and 64 physical education college students with limited soccer experience (considered novice athletes; 32 men and 32 women with matched soccer experience). To assess visual attention, we used a multiple object tracking (MOT) task with four targets among a total of 10 objects moving at a fixed speed of 10°/s in random directions across a computer monitor screen. Tracking accuracy on the MOT task was calculated for each participant as the proportion of correctly selected targets. A univariate analysis of variance was performed, with group (expert, novice) and sex (male, female) as independent variables, and tracking accuracy on the MOT task as the dependent variable to assess whether sports expertise or sex influenced visual attention. Simple effects tests followed by comparisons with Bonferroni corrections were used, and effect size calculations were performed using Cohen's $f$ statistic.

**Results.** Tracking accuracy on the MOT task was significantly affected by sports expertise ($F_{(1,124)} = 91.732$, $p < 0.001$, $\eta_P^2 = 0.425$), with accuracy among expert soccer athletes superior to that among novice soccer athletes. Moreover, a statistically significant interaction between sports expertise and sex was detected ($F_{(1,124)} = 7.046$, $p = 0.009$, $\eta_P^2 = 0.054$). Better tracking performance was observed for male soccer players (mean [SD], 0.39 [0.12]) than for female soccer players (mean [SD], 0.27 [0.08]); $p < 0.01$; $d = 1.17$; $r = 0.51$) but only in the novice group. No significant sex difference was detected in tracking performance between elite male soccer athletes (mean [SD], 0.51 [0.09]) and elite female soccer athletes (mean [SD], 0.49 [0.11]).

**Conclusion.** These findings confirm previous results indicating that long-term extensive sports training develops visual attention as assessed by MOT performance and extend previous findings to include soccer athletes. The findings of a sex difference in visual attention among novice soccer players but not among elite soccer athletes who

![PeerJ logo]

had similar performance requirements and training experience suggest that long-term extensive training may minimize the sex difference in visual attention.

## INTRODUCTION

Visual attention has been well-studied for years and may be one of the best-investigated perception concepts in the field of psychology in the last couple of years. Visual attention can be characterized as the cognitive system that selects a subset of relevant stimuli for further processing (*Harris et al., 2020*). Attention is the dominant factor in cognitive fitness (*Albertella et al., 2023*) as it plays important roles during most daily activities, including driving a car, crossing the street, doing household chores, and participating in sports (*Howard, Uttley & Andrews, 2018*; *Hüttermann & Memmert, 2017*). In sports, athletes must accurately and rapidly extract meaningful visual information to ensure making correct decisions (*Walsh & Vincent, 2014*). This capability is particularly important in open skill sports, such as soccer, in which players must allocate their visual attention to specific components of the soccer field, including to teammates, opponents, and the ball, in order to make good decisions. It is often said that athletes who have the ability to "read the game" (*Romeas, Guldner & Faubert, 2016*) or this "version of the game" (*Faubert & Sidebottom, 2012*) have better sports performance during a game.

Sports scientists have studied the association between sports experience (expertise) and visual attention during the last decade, and those studies have yielded mixed results. A number of studies have demonstrated that expert athletes exhibit better visual attention performance than novice players (*Heppe et al., 2016*; *Voss et al., 2010*; *Wechsler et al., 2021*). It has also been widely demonstrated that extensive sports training may improve cognitive function and may lead to plasticity changes in brain structure and function (*Etnier et al., 2006*; *Kramer & Erickson, 2007*). Research also indicates that long-term physical training has positive transfer effects on relevant perceptual-cognitive functioning (*Heppe et al., 2016*), such as visual attention (*Zhang et al., 2021*). Thus, it is reasonable to suggest that the superior visual attention observed in expert athletes is due to their extensive physical training (*Alves et al., 2013*). However, some studies have detected no significant difference in visual attention performance between experts and novices (*Furley & Memmert, 2010*; *Stothart et al., 2014*). For example, one study (*Memmert, Simons & Grimme, 2009*) showed that the expected differences in the performance of a visual attention task between experts and novice did not appear. Although many studies have examined the association between expertise and visual attention, it remains unclear whether expertise affects visual attention in sports. Thus, the aim of this study was to investigate visual attention in experts and novices to assess whether sports experience affects their visual attention.

Although male athletes and female athletes differ in many aspects of their performance in the field of sports (*Wilmore, 1979*), a sex difference in visual attention has not been

systematically explored in the relevant sports literature. One meta-analytic review found that male athletes exhibit greater cognitive function compared with female athletes (*Voss et al., 2010*). *Lum, Enns & Pratt (2002)* demonstrated that male athletes perform better than female athletes in a special attention task, and *Legault, Sutterlin-Guindon & Faubert (2022)* reported that male athletes exhibit superior performance to female athletes in a visual tracking task. However, the results of another study indicated a lack of a sex difference between male and female volleyball players in the performance of a visual selective attention task (*Alves et al., 2013*). *Notarnicola et al. (2014)* also found that male players show no better performance than female players in a visual-spatial task among volleyball and tennis players. In addition, a recent study by members of our group assessing visual attention found no evidence of a sex difference among basketball players in the performance of a visual attention task (*Jin, Ge & Fan, 2022*). Thus, research examining the effects of sports experience on sex differences in visual attention is still in its early stages and currently lacks compelling evidence.

The multiple object tracking (MOT) task is a powerful and well-established paradigm that has been widely used for investigating visual attention abilities in laboratory research (*Zwierko et al., 2022b*), and numerous studies have shown that it has good reliability and validity (*Cohen, Alvarez & Nakayama, 2011*; *Huang, Mo & Li, 2012*; *Tombu & Seiffert, 2008*). MOT was introduced to perceptual cognitive science more than three decades ago (*Pylyshyn & Storm, 1988*) and has been used to assess several aspects of selective attention (*Meyerhoff & Papenmeier, 2020*), sustained attention (*Koldewyn et al., 2013*), and distributed attention (*Meyerhoff, Papenmeier & Huff, 2017*). In a typical MOT task, participants are asked to track moving targets (initially distinguished by their flashing or changing colors) among distractors in a confined visual scene for a short period of time (*Doran & Hoffman, 2010*). The MOT paradigm well-replicates the visual requirements of team ball sports, for which players need to monitor not only the spatial location of the ball but also the positions and movements of teammates on the court or field (*Mangine et al., 2014*; *Martín et al., 2017*). One behavioral study reported that expert team ball sport athletes showed better MOT task performance compare with novice players (*Faubert, 2013*). However, another study (*Memmert, Simons & Grimme, 2009*) demonstrated that the expected difference in performance in a similar MOT test between expert and novice team ball sport players did not happen. In addition, a sex difference in the performance of a visual attention task among the team ball sport players was doubtful. Thus, there is limited empirical evidence suggesting that male and female team ball sport players differ in visual attention as assessed in the MOT task.

Therefore, the first aim of the current study was to examine the association between sports expertise and visual attention as assessed using a MOT task. We hypothesized that expert athletes would show better tracking performance in the MOT task than novices owing to long term sports training (*Alves et al., 2013*; *Qiu et al., 2018*). The second aim was to explore whether there is a sex difference in visual attention between male athletes and female athletes. We hypothesized that a sex difference in the MOT task would be exhibited only between male athletes and female athletes among novice players, but not among
expert athletes because the extensive sports training of the latter would minimize any sex difference.

## MATERIALS & METHODS

### Participants

*A priori* sample size estimation was conducted using the software G*Power, version 3.1.9, with an effect size of 0.25, $1 - \beta$ of 0.80, and an alpha level of 0.05 (*Cohen, 1992*). This analysis indicated the need for 128 participants in this study, including 64 elite-level soccer athletes (32 men and 32 women) and 64 novice players (32 men and 32 women). The elite athletes were first-level national athletes recruited from college soccer teams and comprised 32 men (mean [SD] age, 22.35 ± 2.31 years; mean [SD] soccer training experience, 10.78 [2.58] years; and mean [SD] training time per week, 13.53 [2.89] h) and 32 women (mean [SD] age, 21.86 [2.47] years; mean [SD] soccer training experience, 9.87 [2.16] years; and mean [SD] training time per week, 12.82 [2.77] h). The novice participants were physical education college students, and comprised 32 men (mean [SD] age, 21.82 [1.71] years) with limited soccer training experience (mean [SD], 2.06 [1.84] years) and training time per week (mean [SD], 2.53 [1.67] h) and 32 women (mean [SD] age, 21.32 [2.28] years), also with limited soccer training experience (mean [SD], 2.87 [2.42] years) and training time per week (mean [SD], 2.79 [1.47] h). All participants reported having normal or corrected-to-normal levels of visual function during testing. None of the participants had experienced a MOT task before or were trained in MOT.

The study protocol was approved by the Ethics Committee of Shanghai University of Sport (approval No. 2015003SUS). All participants provided written informed included descriptions of the study aim, task, procedure, obligations, responsibilities, and rights of participants) prior to the start of the study. Each participant who completed this study received $15 as compensation for their time.

### Stimuli and procedure

The task and stimulus display were controlled using MATLAB R2016a (MathWorks, Natick, MA, USA) and Psychtoolbox 3.0, running on a ThinkPad PC with a 17-inch monitor at a screen resolution of 3,072 × 1,440 and a 120-Hz refresh rate. Participants sat approximately 55 cm from the monitor and tested individually in a quiet and lit room. At the start of each test block, the directive ''press the left mouse button to start the test'' was displayed on the screen. For each trial, a white fixation symbol (+) was first presented for 1,000 ms in the center of a gray background (37.98° × 21.00°), followed by 10 white filled circles (diameter 0.65°) for 1,000 ms. Four of these circles were then highlighted in dark blue and flashed three times for a total of 2 s to indicate their status as targets for the tracking task. After that, the four target circles returned to white so that no clues remained to distinguish them from the non-tracking items (distractors). Thereafter, all 10 filled circles started to move at a fixed speed of 10°/s in random directions, bouncing off the edges of the display frame and having the possibility of crossing one another for an instant while in motion. At 8 s after the start of their motion, all circles stopped moving. The participants were required to press a mouse button to point out the fours targets and

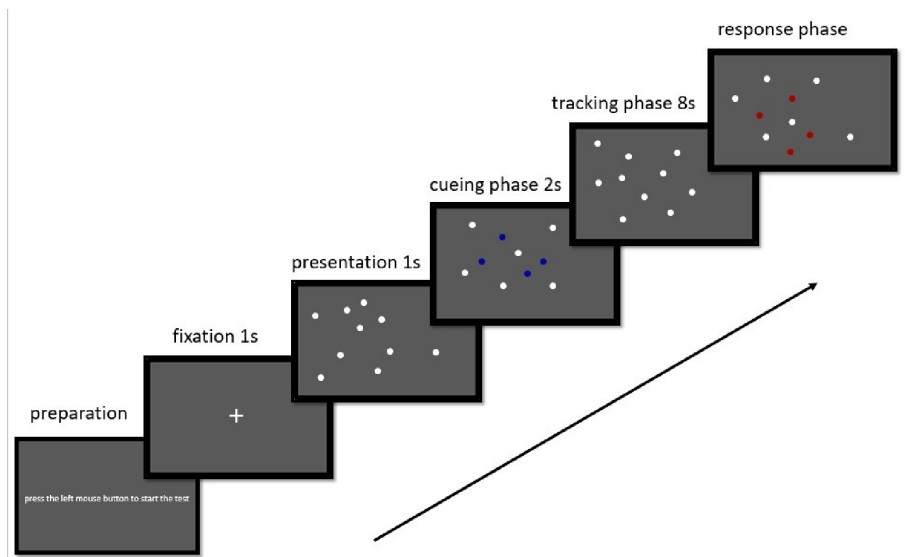

**Figure 1** **Illustration of one trial on the multiple object tracking task.** Ten white filled circles are displayed on the monitor in the presentation phase. At the start of each trial (the cueing phase), four circles turn dark blue and flash three times, for a total of 2 s, to indicate their status as targets for that trial. All 10 white circles then move at 10°/s for 8 s in the tracking phase. When the circles stop moving, participants click with the mouse over the four targets (red circles) during the response phase.

to then trigger the start of the next trial (Fig. 1). The examiner gave no specific instruction about how to perform the MOT task during the test.

The experiment included 40 trials in two blocks, separated by a 4-min rest. The entire task took approximately 20 min to accomplish. Participants were given up to six practice trials to ensure that they were familiarized with the task before the main testing session. To ensure tracking accuracy, we did not require participants to respond quickly (*Qiu et al., 2018*).

## Data and statistical analyses

Raw data were recorded using MATLAB version 9.0.0.341360 (R2016a) software. We calculated tracking accuracy by determining the proportion of correctly picked targets across all 40 trials for each participant. As an example, if a participant identified all four targets 18 of 40 times, the tracking accuracy was 0.45. Statistical analyses were conducted using the SPSS Package for Windows, version 25.0. A univariate analysis of variance (UNIANOVA) was performed, with group (expert, novice) and sex (male, female) as independent variables, and tracking accuracy on the MOT task as the dependent variable. Further analyses were performed using the simple effects test for any significant interaction, and *post-hoc* comparisons were corrected by the Bonferroni procedure. Effect size calculations were performed using Cohen's $f$ statistic, with classifications defined as follows: 0.10, 0.25, and 0.40 were considered small, medium, and large effect sizes, respectively. An alpha level of 0.05 was used to determine statistical significance.

**Table 1  UNIANOVA results for between-subjects effects.**

| Source | SoS-III | df | Mean square | F | p | $\eta_P^2$ |
|---|---|---|---|---|---|---|
| Corrected model | 1.114[a] | 3 | .371 | 38.211 | <.001 | .480 |
| Intercept | 21.846 | 1 | 21.846 | 2,248.851 | <.001 | .948 |
| Sex | .154 | 1 | .154 | 15.854 | <.001 | .113 |
| Group | .891 | 1 | .891 | 91.732 | <.001 | .425 |
| Sex × group | .068 | 1 | .068 | 7.046 | .009 | .054 |
| Error | 1.205 | 124 | .010 | | | |
| Total | 24.164 | 128 | | | | |
| Corrected total | 2.318 | 127 | | | | |

Notes.
SoS-III, , type III sum of squares;  df, , degrees of freedom.
Dependent variable = tracking accuracy.
[a]$R^2 = .480$ (adjusted $R^2 = .468$).

**Table 2  Results for sex and group interactions.**

| | Comparison | | Mean difference | SE | p value[a] | 95% CI | |
|---|---|---|---|---|---|---|---|
| | | | | | | Lower limit | Upper limit |
| *Sex* | Male expert | male novice | .121* | .025 | <.001 | .072 | .169 |
| | Female expert | female novice | .213* | .025 | <.001 | .164 | .262 |
| Group | Expert male | expert female | .023 | .025 | .350 | −.026 | .072 |
| | Novice male | novice female | .116* | .024 | <.001 | .067 | .164 |

Notes.
CI,  confidence interval.
Dependent variable, tracking accuracy
*The mean difference is significant at the .05 level.

# RESULTS

Our UNIANOVA results of tracking accuracy indicated a significant main effect of sex ($F_{(1,124)} = 15.854$, $p < 0.001$, $\eta_P^2 = 0.113$) and of group ($F_{(1,124)} = 91.732$, $p < 0.001$, $\eta_P^2 = 0.425$) and their significant interaction ($F_{(1,124)} = 7.046$, $p = 0.009$, $\eta_P^2 = 0.054$) (Table 1).

Further simple effects analyses by sex showed that tracking accuracy among male experts (mean [SD], 0.51 [0.09]) was significantly better than that among male novice players (mean [SD], 0.39 [0.12]; $p < 0.01$; $d = 1.13$; $r = 0.49$), and that tracking accuracy among female experts (mean [SD], 0.49 [0.11]) was significantly better than that among female novices (mean [SD], 0.27 [0.08]; $p < 0.01$; $d = 2.28$; $r = 0.75$). An analysis by group showed that tracking accuracy among male novices (mean [SD], 0.39 [0.12]) was significantly better than that among female novices (mean [SD], 0.27 [0.08]); $p < 0.01$; $d =1.17$; $r = 0.51$). However, there was no significant difference in tracking accuracy between male experts (mean [SD], 0.51 [0.09]) and female experts (mean [SD], 0.49 [0.11]) (Table 2 and Fig. 2).

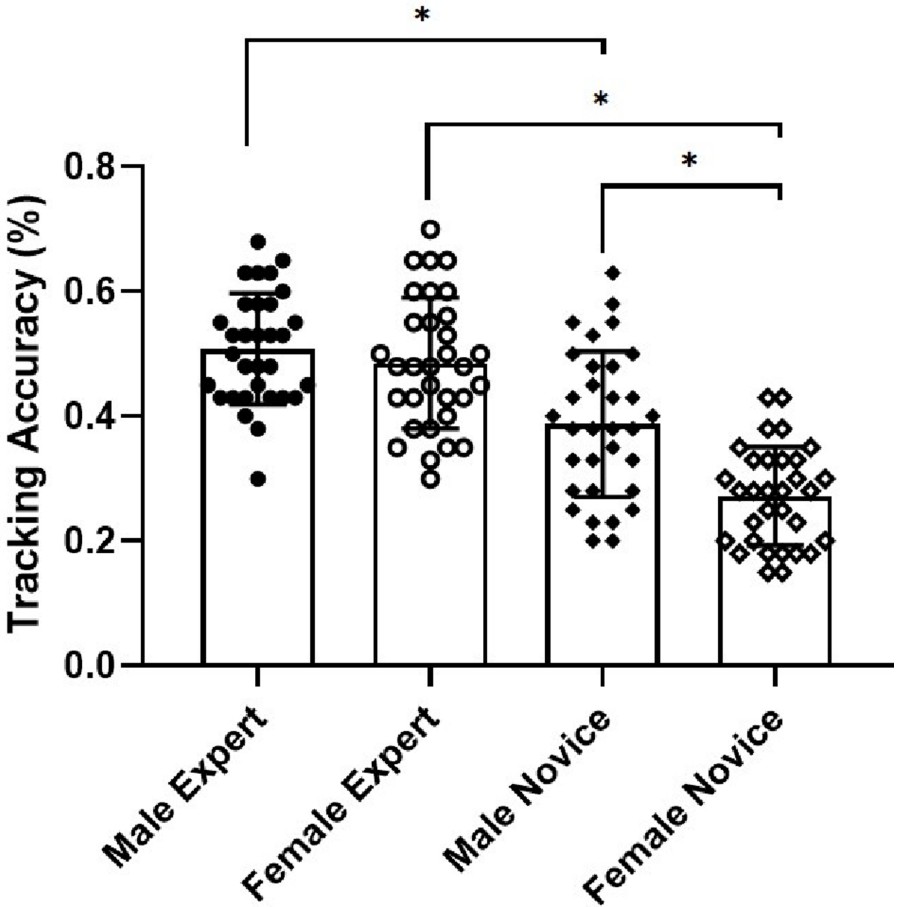

**Figure 2 Comparison of tracking accuracies between groups.** Individual data points indicate the mean tracking accuracy of each participant; the top of each bar, the group mean; and horizontal bars, group standard deviation. $*p < 0.05$ for the indicated comparisons.

## DISCUSSION

The aims of the present study were to (1) examine the association between the level of soccer expertise and visual attention and (2) assess whether a sex difference for visual attention existed among soccer players. Our results provided evidence in support of our first hypothesis that expert soccer athletes would show tracking performance superior to that of novice soccer players in a MOT task. Our results also supported our second hypothesis that visual attention would differ by sex only for the novice group, not for the expert group.

We found that visual attention was associated with the level of sports expertise, with expert soccer athletes showing higher tracking accuracy than novice soccer athletes on the MOT task. This result is consistent with the existing evidence that team ball sport expert athletes have better visual attention than novices (*Meng et al., 2019*; *Verburgh et al., 2014*). A study by *Faubert (2013)* found that a professional team ball sport athlete group showed superior performance to the amateur group on a MOT task. Another study (*Qiu et al.,*

*2018*) showed that elite basketball athletes exhibited better MOT performance than novices. These results are likely because expert soccer athletes benefit from long-term training and competitive experience to fully develop their perceptual-cognitive abilities (*Huijgen et al., 2015*; *Verburgh et al., 2014*). In addition, other studies have reported that sports training has positive transfer effects on visual attention among team ball athletes (*Heppe et al., 2016*; *Zhang et al., 2021*). However, such results are inconsistent with those observed in another study, which found that playing team sports was not correlated with the ability to track multiple objects, even after no less than 10 years of extensive training (*Memmert, Simons & Grimme, 2009*). These discrepant results may be attributable to the use of a lower difficulty MOT task. The study by *Memmert, Simons & Grimme (2009)* used a lower tracking load, with three targets among seven objects—instead of four targets among 10 objects as was used in our study—which is likely not sufficiently sensitive to differentiate this skill among expert athletes and novice players (*Howard, Uttley & Andrews, 2018*). In summary, our results extended those of some previous studies and confirmed our hypothesis that expert athletes would show visual attention as assessed by the MOT task superior to that of novice players.

We also assessed a sex by group interaction. A statistically significant sex difference was detected for MOT task performance in the novice group, whereas male expert soccer athletes did not demonstrate tracking performance superior to expert female soccer athletes. Only a few studies have examined whether engaging in long-term team sports training could minimize sex differences in visual attention. A previous study by *Alves et al. (2013)* reported that male expert volleyball athletes did not show better performance than female expert volleyball athletes on a visual selective attention task. The results of another study also indicated that there was no sex difference for performing a visual spatial task among semi-professional volleyball players, but a sex difference was observed in the control group. Similar results were also reported for no sex difference in visual attention as assessed using a visual test among trained volleyball athletes *Zwierko et al. (2022a)*. A recent study by members of our group (*Jin, Zhao & Zhu, 2023*) reported that men show higher tracking accuracy than women on the MOT task in a group of untrained college students but not among high-level basketball players. These results may be because team sports environments for both male athletes and female athletes require the same perceptual (visual) skills. Given the intensities of competitions in more recent years, the requirements for performance and the opportunities for daily training are relatively similar for both competitive male and female athletes (*Scanlan et al., 2012*; *Senne, 2016*). Other studies have also supported the view that similar long term-training could decrease a sex difference on visual attention (*Notarnicola et al., 2014*; *Ryan, Atkinson & Dunham, 2004*). A second plausible explanation for this finding could be associated with hormonal modulation, with sex hormones having a strong impact on the performance in attention tasks (*Holländer et al., 2005*; *Williams, Barnett & Meck, 1990*). Previous studies have reported that female athletes show a greater advantage from higher androgen levels than male athletes in team sports, such as basketball; thus, androgens may develop this perception in women, but may inhibit it in men (*Holländer et al., 2005*; *Lord & Leonard, 1997*). All these aforementioned results support and may explain our finding of a sex difference in visual attention only

among novice soccer players and not among expert soccer athletes. Nevertheless, there is still much to be learned about the possible cause for this phenomenon. Measurement of androgen levels and the neuropsychological functioning test are necessary to enhance understanding in this area in the future research.

### Limitations

The current study has several limitations. The main limitation was the use of a quasi-experimental research design, rather than a randomized controlled trial although it is not feasible to randomly assign participants to a novice or expert sports group. It is possible that highly practiced soccer players had better tracking of fast objects before they engaged in long-term sports training. Therefore, a future longitudinal study is needed to show changes in visual attention capacity in soccer players over time. One more limitation of the study is that we did not take into account the neurophysiological characteristics that may have influenced the findings. Future studies should include neuropsychological tests to enhance this understanding. Lastly, this study assessed elite soccer players and the results may not be generalizable to athletes engaged in non–ball sports. Despite these limitations, to our knowledge, the present study has the merit of being the first to explore a sex difference for visual attention in soccer players.

## CONCLUSIONS

The results of the present study indicated that expert soccer players exhibited object tracking performance in a MOT task superior to that of novice soccer players. In addition, male soccer athletes displayed superior performance to female soccer athletes only among novice players. No sex difference was observed between male expert soccer athletes and female expert soccer athletes. The findings of this research demonstrate that long-term extensive training in team ball sports, such as soccer, may transfer to the development of visual attention, as assessed by tracking moving objects, and reduce any sex difference. These findings highlight the need to control for sex in future studies that assess visual attention by using MOT task. Ultimately, the findings of this study give applied sport psychology practitioners and coaches valuable information for cultivating future elite athletes, especially female athletes.

### Funding
The authors received no funding for this work.

### Competing Interests
The authors declare there are no competing interests.

### Author Contributions
- Peng Jin conceived and designed the experiments, analyzed the data, prepared figures and/or tables, and approved the final draft.

- Zheqi Ji performed the experiments, prepared figures and/or tables, and approved the final draft.
- Tianyi Wang performed the experiments, authored or reviewed drafts of the article, and approved the final draft.
- Xiaomin Zhu conceived and designed the experiments, authored or reviewed drafts of the article, and approved the final draft.

## Human Ethics

The following information was supplied relating to ethical approvals (i.e., approving body and any reference numbers):

The Ethics Committee of Shanghai University of Sport (approval No. 2015003SUS)

## Data Availability

The raw measurements are available in the Supplementary File.

## Supplemental Information

Supplemental information for this article can be found online at http://dx.doi.org/10.7717/peerj.16286#supplemental-information.

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
