# Peer review of "Association between sports expertise and visual attention in male and female soccer players"

_PeerJ, doi:10.7717/peerj.16286_

## Round 0.1 · original submission · Minor Revisions

I have read the manuscript myself and agree with the reviewers that it is a valid one, as well as I agree that it needs some adjustments to further improve its overall clarity and quality. In this regard, I recommend you to accurately address all the reviewers' comments.

Reviewer 1 ·

Basic reporting

This study examined the impact of soccer expertise level and gender on the capacity for processing an attention-demanding task (MOT) that has some ecological relevance to sports visual processing requirements for team sports. Although in my opinion there is mounting evidence that MOT tasks are better performed in populations with greater team sports expertise, especially in studies with significant sample sizes, there remains a controversy as some studies indicated no differences. Another question still unresolved is the impact of sex differences and its potential interaction with sports expertise levels. What is notable in the present study is that it has the power to resolve these issues for several reasons. It has a substantial sample size, and the groups were perfectly balanced with equal participants in all the four groups included in the study. This allows us to address these two questions relatively clearly.
General comments:
I enjoyed reading the paper as it has clear objectives, it is well written and has significant power to address the objectives. The results are also clear, giving credence to the conclusions.
I believe that this study is of great interest for the sport science community and helps resolve some lingering issues in sports science literature.

Experimental design

The design was fine.

Validity of the findings

Data was provided, statistics and results were sound.

Additional comments

Minor comments:
I found two typos in the paper that should be addressed before publication.
Line 95: Change “that” to “than”
Line 239: Change “showed” to “show”

Reviewer 2 ·

Basic reporting

This study aimed to explore the potential correlation between sports expertise and the development of visual attention using the Multiple Object Tracking (MOT) task. Furthermore, the study aimed to investigate potential disparities in visual attention between male and female athletes. The manuscript is well-composed, exhibiting a consistent structure and clear exposition that enhances its comprehensibility. However, I would like to draw attention to several areas that require further clarification.

Experimental design

1. Regarding attention capacity, the authors employed the MOT task, in which 10 filled circles initiated movement at a constant speed of 10°/s in random directions, rebounding off the edges of the display frame. These circles had the potential to momentarily intersect during motion. After 8 seconds of motion, all circles came to a halt, prompting participants to press a mouse button to identify the four designated targets. Could the authors provide information concerning the psychometric attributes of the utilized tests? For instance, are there reported measures of reliability and validity for these tasks?
2. Did the examiner provide any specific instructions on how to perform the task? Were participants allowed to use any or specific gaze strategy to enhance task performance?
3. Did participants have any prior experience with the MOT task?
4. Were individual eye positions adjusted, and if they were, how was this accomplished?

Validity of the findings

1 When describing the statistical methods, kindly introduce criteria for effect size in post hoc tests.
2. The discussion section presents well-structured content, but it could benefit from an extended focus on the novel findings.
3. It would be valuable to incorporate an explanation of potential mechanisms underlying the observed sex-based differences in attention capacity between male and female athletes, particularly among novice players. What neurophysiological foundations could contribute to such discrepancies?

Additional comments

Please engage in a discourse regarding the generalizability (external validity) of the study's outcomes.

Reviewer 3 ·

Basic reporting

English throughout is solid and quite parsimonious.
The literature focused on the development of athlete attention, its importance in performance and the use of the dependent variable--MOT. Important citations of the MOT literature is mentioned, in particular the work of Faubert and his colleagues at the University of Montreal (where a rather difficult tracking task is used in their research). From the research on attention I recommend the authors visit the work of Robert Nideffer, who has done the best work related to sport expertise. A recent Australian Delphi study by Lucy Albertella et al illustrated a huge consensus that "attention" was the dominant factor in "Cognitive Fitness". I recommend this paper be cited. "Building a transdisciplinaryexpert consensus on the cognitive drivers of performance under pressure: An internationalmulti-panel Delphi study"

I recommend the authors revisit the reporting of their references (Reference page) so that they are all consistent. Currently they are not and I do not know what PeerJ's policy is on this matter.

Results are clearly tied to the hypotheses posed for the study.

Experimental design

Questions asked by this study are important and useful to sport/performance scientists. The findings contribute to our understanding of the development of attention. The quasi experimental design was appropriate for this study, statistical analysis was rigorous, particularly their estimation of sample size necessary for the study. Methodology was described thoroughly with particular emphasis placed on how Multiple Object Tracking (MOT) is measured. As such the study could readily be replicated.

Validity of the findings

The authors interpretation of the findings (in discussion) are solid and do not go beyond actual findings. Conclusions are clearly stated.

Additional comments

No additional comments, other than note my recommendations stated earlier which are "minor" revisions.

---

## Round 0.2 · accepted · Accept

I commend the authors for having accurately addressed all of the reviewers' comments.

Reviewer 2 ·

Basic reporting

No comment

Experimental design

No comment

Validity of the findings

No comment

Additional comments

The authors have addressed all of my comments mentioned in the review. I have no further comments.